# Personalizing Many Decisions with High-Dimensional Covariates

Nima Hamidi[*]        Mohsen Bayati[†]        Kapil Gupta[‡]

## Abstract

We consider the $k$-armed stochastic contextual bandit problem with $d$ dimensional features, when both $k$ and $d$ can be large. To the best of our knowledge, all existing algorithms for this problem have regret bounds that scale as polynomials of degree at least two, in $k$ and $d$. The main contribution of this paper is to introduce and theoretically analyse a new algorithm (REAL-Bandit) with a regret that scales by $r^2(k+d)$ when $r$ is the rank of the $k \times d$ matrix of unknown parameters. REAL-Bandit relies on ideas from low-rank matrix estimation literature and a new row-enhancement subroutine that yields sharper bounds for estimating each row of the parameter matrix that may be of independent interest. We also show via simulations that REAL-Bandit algorithm outperforms existing algorithms that do not leverage the low-rank structure of the problem.

## 1   Introduction

Running online experiments has recently become a popular approach in data-centric enterprises. However, running an experiment involves an opportunity cost or *regret* (e.g., exposing some users to potentially inferior experiences). To reduce this opportunity cost, a growing number of companies leverage multi-armed bandit (MAB) experiments [38, 39, 19] that were initially motivated by the cost of experimentation in clinical trials [41, 27]. Another common feature of online experiments is personalization; users have heterogenous preferences that means the optimal decisions depend on user or product characteristics (also known as *context*). MAB approach for personalizing decisions is therefore called contextual MAB (or contextual bandit) [29]. For example, [30] used contextual bandits to propose a personalized news article recommender system.

There is a large body of literature on algorithms with theoretical guarantees for contextual bandits with linear reward functions. An admittedly incomplete list is [5, 13, 2, 12, 14, 4, 34, 37, 42, 25, 6], and we defer to [7] for additional references. While these papers study the problem under a variety of different assumptions, they can be divided into two groups: (A) when context vectors are arbitrary and can be potentially selected by an adversary, and (B) when context vectors are i.i.d. samples from a fixed (but unknown) probability distribution. Our focus in this paper is the latter group (first studied by [14]). As the number of decisions $T$ (time horizon) grows, the regret bounds for the algorithms in group (A) grow with $\sqrt{T}$. But the algorithms in group (B) take advantage of the i.i.d. assumption and have a significantly lower (logarithmic) dependence in $T$.

Two other important parameters are the number of arms $k$, and the dimension of context vectors $d$. For example, when $d$ grows, the regret bound of [14] grows as $d^3$ which can dominate the dependence on $T$. [6] tackled this difficulty by imposing sparseness assumption and replaced $d^3$ with $s^2$ (up to logarithmic factors) where $s$ is sparsity of the parameter vectors for the reward functions. On the other hand, a careful inspection of the bounds in [14, 6] reveals that their regret bounds could grow by $k^3$ (in the worst case) that can be very large in applications such as assortment optimization [21].

---

[*]Department of Statistics, Stanford University, hamidi@stanford.edu

[†]Graduate School of Business, Stanford University, bayati@stanford.edu

[‡]Airbnb, kapil.gupta@airbnb.com

On the other hand, the lower bounds found in [13] and [14] imply that if there is no structural assumption, the lowest possible regret would grow at least by $kd$. The aim of this paper is to reduce this dependence under a low-rank assumption on the $k \times d$ matrix of parameters of the reward functions. Specifically, each of the $k$ reward functions is represented by a $d$-vector of coefficients (one coefficient per covariate) that is a row of the parameter matrix. This can also be interpreted as imposing a similarity between the reward functions of the $k$ different arms, like in the multi-task learning literature [10]. We propose a new algorithm (called REAL-Bandit) and prove that its regret grows by $r^2(k + d)$ where $r$ is rank of the parameter matrix.

**Contributions.** Our main technical contributions in design and analysis of REAL-Bandit are as follows. *(1) Stronger row-wise guarantees:* To prove regret guarantees for REAL-Bandit, we need bounds for estimating every single row of the matrix. However, existing matrix estimation results provide a bound on the estimation error of the whole matrix, which would be a crude upper bound for the estimation error of each single row. Therefore, REAL-Bandit includes a subroutine (called Row Enhancement) that refines the estimates in order to establish stronger row-wise guarantees that may be of independent interest (see §3 for details). Very recently, [11] provided bounds for the matrix completion problem that are also sharp at the row or entry level, however, their results are only for the matrix completion case and do not apply to our setting. *(2) Implementation:* Our theoretical analysis does not require that each matrix estimation phase in REAL-Bandit is solved to completion. In other words, REAL-Bandit does not need finding a global minimum of the relevant estimator's optimization (penalized maximum likelihood) problem. REAL-Bandit only needs a solution with cost below a certain threshold which can be used to significantly speed up implementation of REAL-Bandit. *(3) Estimator independence:* Over the last decade, several types of estimators have been introduced for recovering low-rank matrices from noisy observations, with varying assumptions and theoretical guarantees. Two of the most common approaches are based on convex optimization [8, 9, 31, 15, 35, 36, 26, 24], or non-convex optimization [40, 22, 23]. Unlike [14, 6] that work with a fixed estimator, REAL-Bandit is designed to be estimator agnostic and works with any matrix estimation algorithm with theoretical guarantees (see §3 for details).

**Other related literature.** A class of decision-making problems with a large number of arms are assortment optimization problems; when a small subset from a potentially large set of items should be selected. Among the rich literature on this topic, [21, 3] are more related to our paper since they consider a dynamic allocation of assortments via multi-armed bandit ideas. [21] (like us) uses low-rank matrix estimation methods for the learning part. However, the problems and models they study are very different. Specifically, they assume that a decision-maker shows a subset of products to a user. Then the user selection is modeled via a multi-nomial logit (MNL) [32] where the parameters of the MNL model form a low-rank matrix with rows representing customer types and columns representing products.

Two other relevant papers are [42, 25] since they too tackle bandit problems with many actions. They introduce algorithms with regrets that scale with spectral dimension of the Laplacian of a graph that has arms as its vertices. These papers are in group (A) of the aforementioned class of bandit papers that are inherently different. Specifically, they assume actions have known feature vectors (with low spectral dimension) that, together with a single unknown parameter vector, define the linear reward functions. There is a reduction from this setting to our problem, only when the action set is allowed to change (see [1]) which is not the case in [42, 25]. Another recent paper in this category is [28]. The main difference between all of these papers and ours, as discussed above, is that we consider i.i.d. context vectors which allows regret bounds that scale logarithmically in $T$ instead of scaling with $\sqrt{T}$. Finally, recent paper of [20] studies a bandit problem where each action is a pair of arms and the reward function is a bilinear function of the feature vectors of each arm, and has a low-rank parameter matrix.

**Organization.** We introduce additional notation in §2. Then the REAL-estimator and REAL-Bandit algorithm are introduced in §3, followed by simulations in §4. In §5 we present our assumptions, statement of the main theorem, as well as its proof. Proofs of lemmas and additional details are deferred to the extended version of the paper [17].

## 2 Setting and notation

Let $\mathbf{B}^\star$ be a $k \times d$ matrix with real-valued entries. We further assume that $\mathbf{B}^\star$ is of rank $r$ with $r \ll \min(k, d)$. At time $t = 1, 2, \cdots$, a *context vector* $X_t \in \mathbb{R}^d$ is drawn from a fixed probability distribution $\mathcal{P}$, independently from $X_s$ for $s < t$. Then, by choosing arm $1 \leq \kappa \leq k$, the reward $y_t := \langle B^\star_\kappa, X_t \rangle + \varepsilon_{\kappa,t}$ is generated, where $B^{\star\top}_\kappa$ is the $\kappa$-th row of the matrix $\mathbf{B}^\star$, and $\varepsilon_{\kappa,t}$ are independent $\sigma^2_\varepsilon$-sub-Gaussian random variables. In addition, $\langle U, V \rangle$ refers to the inner product of vectors $U$ and $V$, and $U^\top$ refers to the transpose of $U$. Throughout, we use bold capital letters for matrices, and use notation $[n]$ for the set $\{1, 2, \ldots, n\}$, when $n$ is an integer. For any two matrices $\mathbf{Y}_1$ and $\mathbf{Y}_2$ with $d$ columns, by $\mathbf{Y}_1 \sqsubseteq \mathbf{Y}_2$, we mean that all rows of $\mathbf{Y}_1$ are also rows of $\mathbf{Y}_2$. Also, for any subset $\mathcal{U}$ of $\mathbb{R}^d$, the notation $\mathcal{P}_\mathcal{U}$ refers to the conditional distribution $\mathcal{P}(\cdot | \mathcal{U})$ of the contexts.

A policy $\pi$, is a sequential decision-making algorithm that, at each time $t$, chooses the arm $\pi_t \in [k]$ given previous observations and the revealed context $X_t$. We will evaluate the performance of a policy by its cumulative regret, defined as $R_T = \sum_{t=1}^T r_t$, where $r_t = \mathbb{E}\big[\max_{\kappa \in [k]} \langle X_t, B^\star_\kappa \rangle - \langle X_t, B^\star_{\pi_t} \rangle\big]$ where expectation is with respect to the randomness of $X_t$, $\varepsilon_t$ and potential randomness introduced by the policy $\pi$. Our goal is to find policies with low $R_T$.

In order to avoid dealing with unnecessary subscripts, for each context vector $X_t \in \mathbb{R}^d$, we define $\mathbf{X}^\pi_t$ to be a $k \times d$ matrix with all elements equal to zero, except for the $\pi_t$-th row which is equal to $X_t^\top$. Using this notation, we have that

$$y_t = \langle \mathbf{B}^\star, \mathbf{X}^\pi_t \rangle + \varepsilon_t, \tag{1}$$

where the inner product for matrices is defined as $\langle \mathbf{U}, \mathbf{V} \rangle := \operatorname{tr}(\mathbf{U}\mathbf{V}^\top)$. Note that $\varepsilon_t$ is actually $\varepsilon_{\pi_t,t}$, but since all noise values are i.i.d., we will drop the dependence on $\pi$.

For a given subset $\mathcal{I} = \{t_1, \ldots, t_n\}$ of $[T]$, consider the set of corresponding context matrices $\{\mathbf{X}^\pi_{t_i} \mid i \in \mathcal{I}\}$. We define an associated *sampling* operator $\mathfrak{X}^\pi_\mathcal{I} : \mathbb{R}^{k \times d} \to \mathbb{R}^n$ to be defined as follows. For any matrix $\mathbf{B} \in \mathbb{R}^{k \times d}$, $\mathfrak{X}^\pi_\mathcal{I}(\mathbf{B})$ is a vector of length $n$ where its $i$-th entry ($i \in [n]$) is given by $[\mathfrak{X}^\pi_\mathcal{I}(\mathbf{B})]_i := \langle \mathbf{B}, \mathbf{X}^\pi_{t_i} \rangle$. Therefore, the vector form of (1) is

$$Y = \mathfrak{X}^\pi_\mathcal{I}(\mathbf{B}^\star) + E,$$

where $E$ is the $n$-vector of all noise values $\varepsilon_{t_1}, \ldots, \varepsilon_{t_n}$. In the remaining, we use the simpler notation $\mathfrak{X}(\cdot)$ instead of $\mathfrak{X}^\pi_\mathcal{I}(\cdot)$ when $\mathcal{I}$ and $\pi$ are implicitly clear.

We also use different norms in our algorithm and analysis in this paper. $\|\cdot\|_2$ refers to the regular $\ell^2$ norm of a vector. The nuclear norm (or trace-norm) of a matrix is denoted by $\|\cdot\|_*$, and $\|\cdot\|_F$ and $\|\cdot\|_\infty$ refer to the Frobenius and the infinity norm of a matrix. Also, for a given distribution $\mathcal{P}$ over $\mathbb{R}^{k \times d}$, we can define the following norm $\|\mathbf{B}\|_\mathcal{P} := \mathbb{E}\big[\langle \mathbf{B}, \mathbf{Z} \rangle^2\big]$ for all $\mathbf{B} \in \mathbb{R}^{k \times d}$ where $\mathbf{Z}$ is drawn from $\mathcal{P}$. Finally, $\|\mathbf{\Gamma}\|_{\infty,2}$ is the maximum of $\|\Gamma_\kappa\|_2$ for $\kappa \in [k]$ (recall that $\Gamma_\kappa^\top$ is $\kappa$-th row of $\mathbf{\Gamma}$). In fact, one of our assumptions that will be stated explicitly later is that the matrix $\mathbf{B}^\star$ belongs to the following set:

$$\mathcal{S} = \{\mathbf{B} \in \mathbb{R}^{k \times d} \mid \|\mathbf{B}\|_{\infty,2} \leq b^\star\},$$

for a positive constat $b^\star$. Also, for a $k$ by $d$ matrix $\mathbf{B}$ of rank $r$ with singular value decomposition $\mathbf{B} = \mathbf{U}\mathbf{D}\mathbf{V}^\top$, we define the *row-incoherence* parameter as

$$\mu(\mathbf{B}) = \sqrt{\frac{k}{r}} \cdot \frac{\|\mathbf{B}\|_{\infty,2}}{\mathbf{D}_{r,r}}. \tag{2}$$

## 3 Algorithm

In this section, we describe the Row-Enhanced and Low-Rank Bandit (REAL-Bandit) algorithm. The algorithm combines ideas from existing literature [14, 6] and a new *row-enhancement* procedure to obtain sharper convergence rate when $k$ is very large. REAL-Bandit algorithm has two disjoint phases for exploration and exploitation, similar to [14, 6]. In the exploration phase, all arms are given an equal chance to be explored to enable the algorithm to obtain an estimate of their corresponding parameters. These *forced-sampling* estimates are *not* sufficiently accurate to pick the best arm with high probability, however, they are accurate enough to rule out all of the arms that are substantially

inferior to the optimal arm. At each time $t$, these estimates are used as a proxy of the actual arm parameters to form a set of *candidate* arms. In order to choose one of these candidates, we need more accurate estimates and so, the algorithm uses the *all-sampling* estimates that are obtained from *all* the observations made thus far to pick the best arm.

However, unlike [14, 6], that estimate each of the arm parameters $\{B_\kappa^\star\}_{\kappa \in [k]}$ separately, our forced-sampling and all-sampling estimates utilize the low-rank assumption on matrix $\mathbf{B}^\star$ and estimate all parameters simultaneously (like in the multi-task learning literature).

**The Estimators.** REAL-Bandit is designed to work with any matrix estimation method that has theoretical guarantees. Two such estimators (developed in the matrix completion literature) are: (1) estimators based on convex optimization and (2) estimators based on non-convex optimization. Before we present these two classes of estimators, we assume that a set of time periods $\mathcal{J} = \{t_1, \ldots, t_n\}$ and their associate observations $(\mathbf{X}_{t_1}^\pi, y_{t_1}), \ldots, (\mathbf{X}_{t_n}^\pi, y_{t_n})$, and a positive constant $\lambda$ are available. We use notations $\bar{\mathbf{B}}(\mathcal{J})$ or $\widehat{\mathbf{B}}(\mathcal{J})$ for estimators of $\mathbf{B}^\star$, that use observations from time periods in $\mathcal{J}$. When $\mathcal{J}$ is clear, we use simpler notations $\bar{\mathbf{B}}$ and $\widehat{\mathbf{B}}$.

*(1) Convex optimization.* In this approach, introduced by [8], the approximation to $\mathbf{B}^\star$ is the minimizer of the following convex program:

$$\text{minimize} \quad n^{-1}\|Y - \mathfrak{X}(\mathbf{B})\|^2 + \lambda\|\mathbf{B}\|_* . \tag{3}$$

In fact, as [16] shows, one just needs a feasible solution $\bar{\mathbf{B}} = \bar{\mathbf{B}}(\mathcal{J}, \lambda)$ that satisfies:

$$n^{-1}\|Y - \mathfrak{X}(\bar{\mathbf{B}})\|^2 + \lambda\|\bar{\mathbf{B}}\|_* \leq n^{-1}\|Y - \mathfrak{X}(\mathbf{B}^\star)\|^2 + \lambda\|\mathbf{B}^\star\|_* . \tag{4}$$

This brings additional flexibility to choose the optimizer and has computational advantages.

*(2) Non-convex optimization.* Another approach is to explicitly impose the low-rank constraint by writing $\mathbf{B}$ as $\mathbf{U}\mathbf{V}^\top$ where $\mathbf{U} \in \mathbb{R}^{k \times r}$ and $\mathbf{V} \in \mathbb{R}^{d \times r}$. The optimization problem would be:

$$\text{minimize}_{\mathbf{U},\mathbf{V}} \quad n^{-1}\|Y - \mathfrak{X}(\mathbf{U}\mathbf{V}^\top)\|^2 + \lambda(\|\mathbf{U}\|_F^2 + \|\mathbf{V}\|_F^2)/2 . \tag{5}$$

One challenge is that this is not a convex program, but it has been shown that under certain conditions, *alternating minimization* can be an effective algorithm [18]. It can also be shown (e.g., see [22, 31]) that minimizing the above loss function is equivalent to solving the following optimization problem:

$$\begin{aligned}\text{minimize} \quad & n^{-1}\|Y - \mathfrak{X}(\mathbf{B})\|^2 + \lambda\|\mathbf{B}\|_* \\ \text{subject to} \quad & \text{rank}(\mathbf{B}) \leq r .\end{aligned} \tag{6}$$

If $\bar{\mathbf{B}}$ is a solution to (6), since $\mathbf{B}^\star$ is also a feasible solution, (4) must hold.

**REAL-estimator.** The existing theory of matrix estimation provides error bounds for $\|\bar{\mathbf{B}} - \mathbf{B}^\star\|_F$ [9, 26, 33, 24, 16]. However, these results do not characterize how this error is distributed across different rows. On the other hand, in order to get a regret bound, we need to control $\|\bar{B}_\kappa - B_\kappa^\star\|_2$ for all $\kappa \in [k]$, and the trivial inequality $\|\bar{B}_\kappa - B_\kappa^\star\|_2 \leq \|\bar{\mathbf{B}} - \mathbf{B}^\star\|_F$ would introduce an unnecessary $\sqrt{k}$ factor. To remedy this, we introduce REAL-estimator that uses a set of *almost* independent observations to improve the row-wise error bound.

As before, let $\mathcal{J} = \{t_1, \ldots, t_n\}$ be a set of $n$ time periods with $t_1 < t_2 < \cdots < t_n$. We split $\mathcal{J}$ to $\mathcal{J}_1 := \{t_1, \ldots, t_{\frac{n}{2}}\}$ and $\mathcal{J}_2 := \{t_{\frac{n}{2}+1}, \ldots, t_n\}$. For any $\mathcal{K} \subseteq [k]$, let $\mathcal{J}^{\mathcal{K}}$ be the subset of $\mathcal{J}$ such that an arm in $\mathcal{K}$ is pulled, i.e. $\pi_{t_i} \in \mathcal{K}$. Moreover, for $\ell \in \{1, 2\}$ and $\mathcal{K} \subseteq [k]$, let $\mathcal{J}_\ell^{\mathcal{K}} := \mathcal{J}^{\mathcal{K}} \bigcap \mathcal{J}_\ell$. For $\kappa \in [k]$, when $\mathcal{K} = \{\kappa\}$, we use superscript $\kappa$ rather than $\{\kappa\}$ for simplicity. Next, for *any* low-rank matrix estimator $\bar{\mathbf{B}}$, Algorithm 1 performs the row-enhancement procedure. We call the output of this algorithm REAL-estimator and denote it by $\widehat{\mathbf{B}}(\mathcal{J})$. The difficulty of analyzing this estimator arises from the fact that the observations are generated in an adaptive fashion, and thus, the results that require independence assumption are not applicable in our case. However, in the analysis we will show that these observations can be *approximated* by i.i.d. samples, and as a result theoretical guarantees can be obtained. In the following, we will state the assumptions formally, and then, we will verify that they continue to hold throughout the analysis.

Before we define the notion of approximately independent, we need a few more notations. For $\kappa \in [k]$, $\mathbf{X}^\kappa$ is a matrix constructed by the set of context vectors of observations for arm $\kappa$, stacked as rows of this matrix. We define $\mathbf{X}_\ell^\kappa$ for $\ell \in \{1, 2\}$ and $\mathcal{J}_\ell$ similarly. Recall that, for matrices $\mathbf{Y}_1$ and $\mathbf{Y}_2$ with $d$ columns, $\mathbf{Y}_1 \sqsubseteq \mathbf{Y}_2$ means that all rows of $\mathbf{Y}_1$ are also rows of $\mathbf{Y}_2$.

---
**Algorithm 1** Row-enhancement procedure

---
**Input:** Low-rank matrix estimator $\bar{\mathbf{B}} \in \mathbb{R}^{k \times d}$, $\mathcal{J} = \{t_1, \ldots, t_n\}$, and observations
  $(\mathbf{X}_{t_1}^\pi, y_{t_1}), \ldots, (\mathbf{X}_{t_n}^\pi, y_{t_n})$.
 1: Initialize, $\widehat{\mathbf{B}} \in \mathbb{R}^{k \times d}$,
 2: Split $\mathcal{J}$ into $\mathcal{J}_1 := \{t_1, \ldots, t_{\frac{n}{2}}\}$ and $\mathcal{J}_2 := \{t_{\frac{n}{2}+1}, \ldots, t_n\}$,
 3: Compute SVD $\bar{\mathbf{B}}(\mathcal{J}_1) = \mathbf{U}\mathbf{D}\mathbf{V}^\top$,
 4: Let $\mathbf{V}_r$ be the matrix containing first $r$ columns of $\mathbf{V}$,
 5: **for** $\kappa = 1, 2, \cdots, k$ **do**
 6:   Let $\hat{\vartheta}_\kappa = \arg\min_{\vartheta \in \mathbb{R}^r} \sum_{t_i \in \mathcal{J}_2^\kappa} (y_{t_i} - \langle \mathbf{V}_r \vartheta, X_{t_i} \rangle)^2$,
 7:   Set row $\kappa$ of $\widehat{\mathbf{B}}$ to $(\mathbf{V}_r \hat{\vartheta}_\kappa)^\top$.
 8: **end for**
 9: Return $\widehat{\mathbf{B}}$.

---

**Definition 3.1** (Approximately independence). Let $\mathcal{J}$ be a given set of $n$ time periods and $\mathcal{P}$, $\mathcal{P}_\mathcal{U}$, and $\mathcal{P}_\mathcal{V}$ be three distributions. Then, for $\kappa \in [k]$, we say that $\mathcal{J}^\kappa$ is a $(n_\mathcal{U}, n_\mathcal{V})$-*approximately independent* set of observations if there exists random matrices $\mathbf{X}_\mathcal{U}^\kappa$ and $\mathbf{X}_\mathcal{V}^\kappa$ such that

1. $\mathbf{X}_\mathcal{U}^\kappa \sqsubseteq \mathbf{X}^\kappa$ and $\mathbf{X}^\kappa \sqsubseteq \mathbf{X}_\mathcal{V}^\kappa$,

2. All rows of $\mathbf{X}_\mathcal{U}^\kappa$ are independent samples of $\mathcal{P}_\mathcal{U}$,

3. All rows of $\mathbf{X}_\mathcal{V}^\kappa$ are independent samples from either $\mathcal{P}$ or $\mathcal{P}_\mathcal{V}$,

4. $\mathbf{X}_\mathcal{U}^\kappa$ and $\mathbf{X}_\mathcal{V}^\kappa$ have $n_\mathcal{U}$ and $n_\mathcal{V}$ rows respectively.

This definition requires the observations for a row $\kappa$ to lie between two sets of i.i.d. samples. This notion becomes extremely useful whenever one can prove that $n_\mathcal{U}$ and $n_\mathcal{V}$ are of the same order.

Next, we specify the conditions that $\mathcal{P}$, $\mathcal{P}_\mathcal{U}$, and $\mathcal{P}_\mathcal{V}$ need to meet so that we can prove error bounds.

**Definition 3.2.** We say that a distribution $\mathcal{P}(\cdot)$ on $\mathbb{R}^d$ is $(\gamma_{\min}, \gamma_{\max}, \sigma_X)$-diverse if

$$\gamma_{\min} \leq \lambda_{\min}(\mathbf{\Sigma}) \leq \lambda_{\max}(\mathbf{\Sigma}) \leq \gamma_{\max},$$

where $\mathbf{\Sigma} = \mathbb{E}[XX^\top]$, and $\mathbf{\Sigma}^{-\frac{1}{2}}X$ is $\sigma_X^2$-sub-Gaussian (i.e., for any deterministic unit vector $u \in \mathbb{R}^d$, the real-valued random variable $u^\top \mathbf{\Sigma}^{-\frac{1}{2}}X$ is $\sigma_X^2$-sub-Gaussian).

We will treat $\sigma_X$ as a constant. Note that, for instance, when $X$ follows a multivariate Gaussian distribution, then $\sigma_X = 1$.

In our proofs, we will show that whenever $\mathcal{J}$ can be split into two *almost* independent halves, then row-enhancement procedure gives us sharper per-row guarantees than the raw matrix estimator $\bar{\mathbf{B}}$.

**The REAL-Bandit algorithm.** Here, we describe REAL-Bandit algorithm presented in Algorithm 2. As mentioned earlier, this algorithm has disjoint exploration and exploitation phases which are specified by a *force-sampling rule* $f : \mathbb{N} \to [k] \cup \{\varnothing\}$. At time $t$, the force-sampling rule decides between *forcing* the arm $f_t \in [k]$ to be pulled or *exploiting* the past data, indicated by $f_t = \varnothing$. By $\mathcal{F}_t$, we denote the time periods that an arm was forced to be pulled, i.e. $\mathcal{F}_t := \{\tau \leq t : f_\tau \in [k]\}$. For simplicity, we also use $\mathcal{A}_t := [t]$ to refer to the all time periods up to time $t$. The force-sampling rule that we use is a randomized function that picks an arm $\kappa \in [k]$ with probability

$$\mathbb{P}(f_t = \kappa) = \begin{cases} \frac{1}{k} & \text{if } t \leq 2\rho\log(\rho), \\ \frac{\rho}{k[t - \rho\log(\rho) + 1]} & \text{if } t > 2\rho\log(\rho), \end{cases} \quad (7)$$

and $f_t = \varnothing$ otherwise. We will specify the hyper-parameter $\rho$ in §5. As we will see in our analysis, this force-sampling rule ensures that $\mathcal{F}_t^\kappa$ grows as $\mathcal{O}(\log t)$ for all $\kappa \in [k]$. One can alternatively use any force-sampling rule that has this rate of exploration.

**Remark 1.** *The algorithm proposed in [14, 6] are similar to the REAL-Bandit. They, however, use a deterministic force-sampling rule (that can be used here as well). However, our randomized rule brings practical advantages in exchange for a slightly more complex theoretical analysis.*

Now, let $\bar{\mathbf{B}}^F$ and $\bar{\mathbf{B}}^A$ be two low-rank matrix estimators (obtained from observations of the force-sampling rounds and the all-sampling rounds respectively) and denote by $\widehat{\mathbf{B}}^F$ and $\widehat{\mathbf{B}}^A$ their corresponding REAL-estimators, introduced above. These estimators serve different purposes in our algorithm. We will show that the forced-samples estimator $\widehat{\mathbf{B}}^F$ satisfies $\left\| \widehat{B}_\kappa^F - B_\kappa^\star \right\|_2 \leq \mathcal{O}(1)$ with probability at least $1 - \mathcal{O}(1/t)$ for *all* arms $\kappa \in [k]$. The key idea is that $\mathcal{O}(\log t)$ i.i.d. samples are enough to get such a guarantee. These estimates are then only used to rule out some arms that are *very* far from the optimal arm. The threshold for eliminating sub-optimal arms is determined by a hyper-parameter $h$ that is given to the algorithm. This parameter can be thought of as the *average gap* of the problem.

The remaining arms are *candidates* of being the optimal arm. Then, the all-samples estimator $\widehat{\mathbf{B}}^A$ comes into play. This estimator is used to pick the *best* arm among these candidate arms. We will show that $\widehat{\mathbf{B}}^A$ enjoys the sharper bound $\left\| \widehat{B}_\kappa^A - B_\kappa^\star \right\|_2 \leq \mathcal{O}\left(1/\sqrt{t}\right)$ for *all optimal* arms $\kappa \in \mathcal{K}_{\text{opt}} \subseteq [k]$ with probability at least $1 - \mathcal{O}(1/t)$ where $\mathcal{K}_{\text{opt}}$ is defined formally in Assumption 3 of §5. This sharper rate improves the accuracy of the decisions made by the algorithm significantly.

---

**Algorithm 2** REAL-Bandit algorithm

---

**Input:** Force-sampling rule $f$, gap $h$.
1: **for** $t = 1, 2, \cdots$ **do**
2:     Observe $X_t \sim \mathcal{P}$,
3:     **if** $f_t \neq 0$ **then**
4:        $\pi_t \leftarrow f_t$
5:     **else**
6:        $\mathcal{C} = \left\{ \kappa \in [k] \mid \left\langle X_t, \widehat{B}_\kappa^F(\mathcal{F}_{t-1}) \right\rangle \geq \max_{\ell \in [k]} \left\langle X_t, \widehat{B}_\ell^F(\mathcal{F}_{t-1}) \right\rangle - \frac{h}{2} \cdot \|X_t\|_2 \right\}$
7:        $\pi_t \leftarrow \arg\max_{\kappa \in \mathcal{C}} \left\langle X_t, \widehat{B}_\kappa^A(\mathcal{A}_{t-1}) \right\rangle$
8:     **end if**
9: **end for**

---

## 4   Simulations

We compared the REAL-Bandit algorithm with four other algorithms: OLS-Bandit of [14], but we use the improved version from [6] that filters sub-optimal arms, LASSO-Bandit of [6], OFUL of [2] which is based on the Upper Confidence Bound (UCB) idea, and Thompson sampling (the version from [37]). Taking $k = 201, d = 200$, and $r = 3$, we generated matrix $\mathbf{B}^\star$ as $\mathbf{U}\mathbf{V}^\top$ where rows of $\mathbf{U} \in \mathbb{R}^{201 \times 3}$ and $\mathbf{V} \in \mathbb{R}^{200 \times 3}$ are drawn independently and uniformly from the unit sphere in $\mathbb{R}^3$. Noise variance is 1 and features are i.i.d. $\mathcal{N}(\mathbf{0}, \mathbf{I}_d)$. We gave Thompson sampling the true prior mean and variance of the arm parameters, and the true noise variance. Similarly, OFUL had access to the true noise variance. Other parameters of OLS-Bandit, LASSO-Bandit, and OFUL are selected as in [6]. We generated 10 data sets and executed all algorithms for a time horizon of length $T = 40,000$. Figure 1 shows average cumulative regret (with 1 SE error bars) for all algorithms across these 10 runs.. The results of this simulation support our theoretical analysis, that REAL-Bandit takes advantage of the low-rank structure of the problem parameters and significantly outperforms other benchmarks that do not leverage the structure.

## 5   Analysis

This section is dedicated to the analysis of REAL-Bandit. We will first state the assumptions underlying the analysis and then state the main theorem of this section. A discussion of some of these assumptions can be found in [6].

**Assumption 1** (Parameter set). Assume the rank of $\mathbf{B}^\star$ is $r$, $\|\mathbf{B}^\star\|_{\infty,2} \leq b^\star$, and $\mu^\star := \mu(\mathbf{B}^\star)$ where $\mu(\cdot)$ is defined in (2).

**Assumption 2** (Margin condition). For any $a > 0$, there is a constant $c_0 > 0$ such that $\mathbb{E}[N_a] \leq k c_0 a$, where the random variable $N_a$ is defined by $N_a := \sum_{\kappa=1}^k \mathbb{I}\left( \left\langle X, B_{\kappa^*}^\star - B_\kappa^\star \right\rangle \leq a \cdot b^\star \cdot \|X\|_2 \right)$ where $\kappa^* = \kappa^*(X)$ is the optimal arm, given context vector $X$.

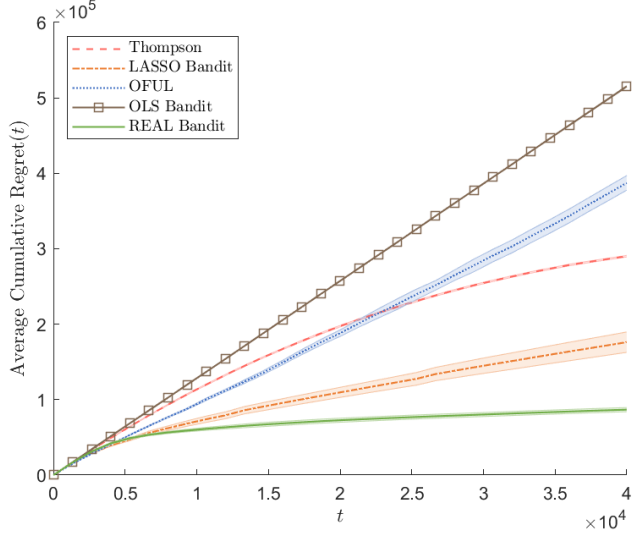

Figure 1: Cumulative regret of REAL-Bandit versus LASSO-Bandit, OFUL, OLS-Bandit, and Thompson sampling for $(k, d, r) = (201, 200, 3)$.

**Assumption 3** (Arm optimality). Let $\mathcal{K}_{opt}$ and $\mathcal{K}_{sub}$ be a partitioning of $[k]$. Then, for some $h > 0$, the following conditions hold:
1) For any sub-optimal arm $\kappa \in \mathcal{K}_{sub}$, $\langle X, B_\kappa^\star \rangle \leq \max_{\kappa'} \langle X, B_{\kappa'}^\star \rangle - h \cdot \|X\|_2$ for any context $X$,
2) For each arm $\kappa \in \mathcal{K}_{opt}$, where $\mathbb{P}(X \in \mathcal{U}_k) \geq \frac{p^*}{|\mathcal{K}_{opt}|}$, where $\mathcal{U}_\kappa$ is defined by

$$\mathcal{U}_\kappa := \left\{ X \in \mathbb{R}^d \mid \langle X, B_\kappa^\star \rangle > \max_{\kappa' \neq \kappa} \langle X, B_{\kappa'}^\star \rangle + h \cdot \|X\|_2 \right\},$$

3) For each arm $\kappa \in \mathcal{K}_{opt}$, there exists constant $q^* > 0$ such that $\max_{\kappa \in \mathcal{K}_{opt}} \mathbb{P}(X \in \mathcal{V}_\kappa) \leq \frac{q^*}{|\mathcal{K}_{opt}|}$ where the set $\mathcal{V}_\kappa$ is defined as

$$\mathcal{V}_\kappa := \left\{ X \in \mathbb{R}^d \mid \langle X, B_\kappa^\star \rangle > \max_{\kappa' \neq \kappa} \langle X, B_{\kappa'}^\star \rangle - h \cdot \|X\|_2 \right\}.$$

**Assumption 4** (Diversity). For all $\kappa \in \mathcal{K}_{opt}$, $\mathcal{P}$, $\mathcal{P}_{\mathcal{U}_\kappa}$, and $\mathcal{P}_{\mathcal{V}_\kappa}$ are $(\gamma_{\min}, \gamma_{\max}, \sigma_X)$-diverse.

**Assumption 5** (Low-rank estimators). Assume the following tail bounds hold for $\bar{\mathbf{B}}^F$ and $\bar{\mathbf{B}}^A$:
1) Let $\mathcal{J}$ be a set of $n$ time periods such that for each arm $\kappa \in [k]$, the matrix $\mathbf{X}^\kappa$ of the context vectors associated to the arm $\kappa$ has i.i.d. rows sampled from $\mathcal{P}$. Then,

$$\mathbb{P}\big(\big\|\bar{\mathbf{B}}^F(\mathcal{J}) - \mathbf{B}^\star\big\|_F \geq \delta\big) \leq \exp\left(-\frac{c_1 \delta^2 n}{(k+d)r}\right),$$

holds when $\frac{n}{\log(n)} \geq c_2(1 + \frac{1}{\delta^2})r(k+d)$, $\delta := \sqrt{\frac{\gamma_{\min}}{\gamma_{\max}}} \cdot \sqrt{\frac{k}{r}} \cdot \frac{h}{64\mu^\star}$, and $c_1, c_2$ are positive constants.

2) Let $\mathcal{J}$ be a set of $n$ observations, such that for all $\kappa \in \mathcal{K}_{opt}$, $\mathcal{J}^\kappa$ is a set of $(\frac{np^*}{2}, 2nq^*)$-approximately independent observations. Then, we get

$$\mathbb{P}\left(\big\|\Pi_{opt}[\bar{\mathbf{B}}^A(\mathcal{J}) - \mathbf{B}^\star]\big\|_F \geq \left[\frac{\sqrt{d}\sigma_\varepsilon \vee b^\star}{\sqrt{d}\gamma_{\min}}\right] \cdot \sqrt{\frac{c_3 r(k+d)\log(n)}{np^*}}\right) \leq \frac{1}{n},$$

provided that $(n/\log n) \geq c_2 r(k+d)$ for some constant $c_3 > 0$, and $\Pi_{opt} : \mathbb{R}^{k \times d} \to \mathbb{R}^{k \times d}$ denotes the linear function that sets the rows corresponding to the sub-optimal arms to zero and keeps the rest unchanged.

Now, we are prepared to state our main theoretical result.

**Theorem 1.** *If Assumptions 1-5 hold, then the cumulative regret of Algorithm 2 is bounded above by*

$$\frac{R_T}{x_\mathfrak{n}} \leq C \left[ c_2 b^\star (1 + \frac{1}{\delta^2}) r(k+d) \log(T) \right] + C' \left[ \frac{c_0 r^2 (k+d) \log(T)^2}{b^\star p^\star} \right],$$

*where $C > 0$ is a constant, the forced-sampling parameter $\rho$ is set to $2c_2(1 + \delta^{-2})r(k+d)$, and*

$$C' := c_3 \mu^{\star 2} \cdot \frac{\gamma_{\max}^2 q^\star}{\gamma_{\min}^2 p^\star} \cdot \left( \frac{d\sigma_\varepsilon^2 \vee b^{\star 2}}{d\gamma_{\min}} \right), \quad x_\mathfrak{n} := \sup_{\|V\|_2 = 1} \mathbb{E} \left[ \|X\|_2 \mid X = \|X\|_2 V \right].$$

Before describing the proof of Theorem 1, we state four key lemmas that will be used in the proof. Due to space limitations, we defer proofs of the lemmas to the extended version of the paper [17].

**Lemma 1.** *The force-sampling sets created by the force-sampling rule* (7) *satisfy the following inequalities, for all $t \geq 2\rho \log(\rho)$, provided that $\rho \geq 24$,*

$$\mathbb{P} \left( |\mathcal{F}_t| \geq 6\rho \log t \right) \leq t^{-1} \quad \text{and} \quad \mathbb{P}(|\mathcal{F}_t| \leq [\rho/2] \log t) \leq t^{-3}.$$

**Lemma 2.** *Let $\mathcal{I}$ be a (deterministic) subset of the forced-sampling observations and by $\mathcal{I}_\kappa \subseteq \mathcal{I}$, we denote the observations corresponding to arm $\kappa \in [k]$. Then, the following inequality holds,*

$$\mathbb{P} \left( \frac{|\mathcal{I}_\kappa|}{|\mathcal{I}|} \leq \frac{1}{2k} \,\bigg|\, |\mathcal{I}| \right) \leq \exp \left[ -\frac{|\mathcal{I}|}{8k} \right].$$

**Lemma 3.** *For all $t \geq 10c_2(1 + \delta^{-2})r(k+d) \log(kd)$ and $\kappa \in [k]$, with probability at least $1 - 10t^{-3}$, the following inequality holds*

$$\left\| \widehat{B}_\kappa^F - B_\kappa^\star \right\|_2 \leq h/4.$$

**Lemma 4.** *For all $t > 10c_2 r(k+d) \log(kd)$ and $\kappa \in \mathcal{K}_{opt}$, with probability at least $1 - 100r(k+d)t^{-1}$, we have*

$$\left\| \widehat{B}_\kappa^A - B_\kappa^\star \right\|_2 \leq 10 \sqrt{\frac{C' r^2 (k+d) \log(t)}{kp^\star t}}.$$

*Proof of Theorem 1.* Following the lines of the proof of Theorem 1 in [6], we define $G(\cdot)$ as

$$G(\mathcal{F}_t) := \begin{cases} 1 & \text{if } \|\widehat{B}_\kappa^F(\mathcal{F}_{t-1}) - B_\kappa^\star\|_2 \leq \frac{h}{4} \text{ for all } \kappa \in [k], \\ 0 & \text{otherwise.} \end{cases}$$

Define $c_4 := 10c_2(1 + \delta^{-2})r(k+d) \log(kd)$. Then, we split the regret of the algorithm into the following three cases and bound each case separately:
(a) Initialization (i.e., when $t \leq c_4$) and forced-sampling rounds.
(b) When $t > c_4$ and $G(\mathcal{F}_{t-1}) = 0$.
(c) When $t > c_4$ and $G(\mathcal{F}_{t-1}) = 1$, but a suboptimal arm is chosen due to inaccurate all-sampling estimates.

Let $R_T^{(a)}$, $R_T^{(b)}$, and $R_T^{(c)}$ denote the regret incurred in the above cases, respectively. Clearly, we have that $R_T = R_T^{(a)} + R_T^{(b)} + R_T^{(c)}$.

Before proving upper bounds, note that, for each suboptimal choice, the regret incurred at each step is at most $\langle X, B_\kappa^\star - B_{\kappa'}^\star \rangle$ for some $\kappa, \kappa'$ which in turn is bounded above by

$$|\langle X, B_\kappa^\star - B_{\kappa'}^\star \rangle| \leq \|X\|_2 \cdot \|B_\kappa^\star - B_{\kappa'}^\star\|_2 \leq 2b^\star \cdot \|X\|_2.$$

This fact can be used to obtain regret bounds by bounding the number of times that each suboptimal arm is pulled. Clearly, for part (a), this number is less than or equal to $c_4 + |\mathcal{F}_T|$. Using Lemma 1,

$$\mathbb{E} \left[ R_T^{(a)} \right] \leq 2b^\star x_\mathfrak{n} \mathbb{E}[c_4 + |\mathcal{F}_T|] \leq 2b^\star x_\mathfrak{n} \left( c_4 + 6\rho \log T \right).$$

Next, it follows from the definition of $G(\mathcal{F}_{t-1})$ and Lemma 3 that the number of times that $G(\mathcal{F}_{t-1}) = 0$ is controlled by

$$\mathbb{E}\left[\sum_{t=c_4+1}^{T} [1 - G(\mathcal{F}_{t-1})]\right] = \sum_{t=c_4+1}^{T} \mathbb{P}(G(\mathcal{F}_{t-1}) = 0) \leq \sum_{t=c_4+1}^{T} 10t^{-3} \leq \sum_{t=c_4+1}^{T} 10t^{-1} \leq 10\log(T).$$

Now, since $R_T^{(b)} \leq 2b^\star \mathbb{E}\left[\sum_{t=c_4+1}^{T} \|X_t\|_2 \cdot [1 - G(\mathcal{F}_{t-1})]\right]$, we have

$$R_T^{(b)} \leq 2b^\star x_{\mathfrak{n}} \mathbb{E}\left[\sum_{t=c_4+1}^{T} (1 - G(\mathcal{F}_{t-1}))\right] \leq 20b^\star x_{\mathfrak{n}} \log(T).$$

Finally, we need to find an upper bound for $R_T^{(c)}$. It follows from a slightly modified version of Lemma EC.18 in [6] that, whenever $G(\mathcal{F}_{t-1}) = 1$, the set $\mathcal{C}$ contains the optimal arm and no suboptimal arm. In particular, if the best arm is $\kappa^*$, we get the following inequality for all $\kappa \in \mathcal{C}$

$$0 < \langle X_t, B_{\kappa^*}^\star - B_\kappa^\star \rangle < h \cdot \|X_t\|_2. \tag{8}$$

Therefore, whenever $G(\mathcal{F}_{t-1}) = 1$, for any $\kappa \in \mathcal{C}$, we have $X \in \mathcal{V}_\kappa$ and if $X \in \mathcal{U}_\kappa$, then $\mathcal{C} = \{\kappa\}$. Now, we are ready to use Lemma 4 to bound the probability of pulling an incorrect arm. Letting

$$\mathcal{E}_t^A := \left\{ \exists \kappa \in \mathcal{K}_{opt} : \|\widehat{B}_\kappa^A(\mathcal{A}_t) - B_\kappa^\star\|_2 > \sqrt{\frac{c_5}{t}} \right\},$$

where $c_5 := \frac{100 C' r^2 (k+d)\log(T)}{kp^*}$. Now, using Lemma 4, we have for all $t > c_4$,

$$\mathbb{P}(\mathcal{E}_t^A) \leq \frac{100(k+d)r}{t}. \tag{9}$$

Now, recall $\kappa^*$ denotes the optimal arm and the arm $\pi_t$ is the pulled arm. For (random variable) $\kappa \in [k]$, define $\mathcal{D}_\kappa := \left\{ \langle X_t, B_{\kappa^*}^\star - B_\kappa^\star \rangle \geq 2\sqrt{\frac{c_5}{t}} \cdot \|X_t\|_2 \right\}$. It follows from the definition of $r_t$ that

$$r_t = \mathbb{E}\left[\langle X_t, B_{\kappa^*}^\star - B_{\pi_t}^\star \rangle\right] \leq \mathbb{E}\left[\langle X_t, B_{\kappa^*}^\star - B_{\pi_t}^\star \rangle \mathbb{I}\left(\mathcal{D}_{\pi_t} \cup \mathcal{E}_t^A\right)\right] + \mathbb{E}\left[\langle X_t, B_{\kappa^*}^\star - B_{\pi_t}^\star \rangle \mathbb{I}\left(\mathcal{D}_{\pi_t}^c \cap \mathcal{E}_t^{Ac}\right)\right]$$

$$\leq \mathbb{E}\left[\langle X_t, B_{\kappa^*}^\star - B_{\pi_t}^\star \rangle \mathbb{I}\left(\left\{\langle X_t, \widehat{B}_{\pi_t}^A \rangle \geq \langle X_t, \widehat{B}_{\kappa^*}^A \rangle\right\} \cap \left(\mathcal{D}_{\pi_t} \cup \mathcal{E}_t^A\right)\right)\right]$$

$$\quad + \mathbb{E}\left[\langle X_t, B_{\kappa^*}^\star - B_{\pi_t}^\star \rangle \mathbb{I}\left(\mathcal{D}_{\pi_t}^c \cap \mathcal{E}_t^{Ac}\right)\right]$$

$$\leq 2x_{\mathfrak{n}}\left[b^\star \mathbb{P}\left(\left\{\langle X_t, \widehat{B}_{\pi_t}^A \rangle \geq \langle X_t, \widehat{B}_{\kappa^*}^A \rangle\right\} \cap \left(\mathcal{D}_{\pi_t} \cup \mathcal{E}_t^A\right)\right) + \sqrt{\frac{c_5}{t}} \mathbb{P}\left(\mathcal{D}_{\pi_t}^c \cap \mathcal{E}_t^{Ac}\right)\right]. \tag{10}$$

Note that $\langle X_t, \widehat{B}_{\pi_t}^A \rangle \geq \langle X_t, \widehat{B}_{\kappa^*}^A \rangle$ in combination with the definition of $\mathcal{D}_{\pi_t}$ implies that

$$0 \geq \left\langle X_t, \widehat{B}_{\kappa^*}^A - B_{\kappa^*}^\star \right\rangle + \left\langle X_t, B_{\pi_t}^\star - \widehat{B}_{\pi_t}^A \right\rangle + 2\sqrt{\frac{c_5}{t}} \cdot \|X_t\|_2.$$

And this entails that at least one of the following inequalities hold:

$$\|X_t\|_2 \cdot \|B_{\kappa^*}^\star - \widehat{B}_{\kappa^*}^A\|_2 \geq \left|\left\langle X_t, B_{\kappa^*}^\star - \widehat{B}_{\kappa^*}^A \right\rangle\right| \geq \sqrt{\frac{c_5}{t}} \cdot \|X_t\|_2$$

$$\|X_t\|_2 \cdot \|\widehat{B}_{\pi_t}^A - B_{\pi_t}^\star\|_2 \geq \left|\left\langle X_t, \widehat{B}_{\pi_t}^A - B_{\pi_t}^\star \right\rangle\right| \geq \sqrt{\frac{c_5}{t}} \cdot \|X_t\|_2.$$

Since $\mathcal{C}$ does not contain any suboptimal arm, we have that $\left\{\langle X_t, \widehat{B}_{\pi_t}^A \rangle \geq \langle X_t, \widehat{B}_{\kappa^*}^A \rangle\right\} \bigcap \mathcal{D}_{\pi_t} \subseteq \mathcal{E}_t^A$. This fact, combined with (9) means for all $t > c_4$, the following holds

$$\mathbb{P}\left(\left\{\langle X_t, \widehat{B}_{\pi_t}^A \rangle \geq \langle X_t, \widehat{B}_{\kappa^*}^A \rangle\right\} \cap \left(\mathcal{D}_{\pi_t} \cup \mathcal{E}_t^A\right)\right) \leq \frac{100(k+d)r}{t}.$$

Finally, by using the margin condition, we get that

$$\mathbb{P}\left(\mathcal{D}_{\pi_t}^c \cap \mathcal{E}_t^{Ac}\right) \leq \mathbb{P}\left(\mathcal{D}_{\pi_t}^c\right) \leq \mathbb{E}\left[N_{\frac{2}{b^\star}\sqrt{\frac{c_5}{t}}}\right] \leq kc_0 \frac{2}{b^\star}\sqrt{\frac{c_5}{t}}.$$

Therefore, using (10), we have

$$R_T^{(c)} \leq \sum_{t=c_4+1}^{T} 2x_{\mathfrak{n}}\left[\frac{100 \cdot r(k+d) \cdot b^\star + kc_0c_5}{t}\right] \leq 2x_{\mathfrak{n}} \cdot \left[100 \cdot r(k+d) \cdot b^\star + \frac{4kc_0c_5}{b^\star}\right]\log(T).$$

$\square$

**Acknowledgments**

The authors gratefully acknowledge support of the National Science Foundation (CAREER award CMMI: 1554140), Stanford Data Science Initiative, and Human-Centered AI Initiative.

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
