[Reviews · NeurIPS 2019]

Reviewer 1



Summary: This paper studies the problem of regret minimization for low-rank linear contextual bandits. In the setting considered, there are k arms, each with an unknown d dimensional vector where k and d are assumed to be large. However, the k by d matrix of arms’ feature vectors is assumed to be low rank with rank r, making some amount of information sharing for estimating different arms possible. At each time step, the algorithm proceeds as follows. First, a context is drawn iid from a distribution, similar to existing work. The algorithm uses an estimate of the k by d matrix (referred to as B) to select the optimal arm to pull and observes a reward which is the inner product of the selected arm and the context with some additive, sub-Gaussian noise. The objective of the algorithm is to minimize cumulative regret. It achieves this via a two phase algorithm, where some initial random exploration builds an estimate of B used to toss away definitely suboptimal arms, and then an exploitation phase for the remainder of the arms. The paper achieves regret scaling like r^2(k+d)log(T), better that the sqrt(kd) log(T) regret necessary without a low rank assumption. Strengths: The paper is generally well written, and the first three sections are well laid out. The flow is logical and shows how to construct tighter estimates of individual arms, and then just plugs this result straight in. Furthermore, the result is a strong one and it can easily be seen the effect of the rank both on the algorithm itself and the resulting bound. Additionally, the row-wise estimation bounds are of separate interest and could have impact in other problems outside the bandit setting. All in all, a fun and interesting paper to read. Questions and comments: - The authors mention the issue of having different numbers of samples for different rows due to the adaptive nature of the algorithm, and note that this breaks down guarantees for one of the estimates of B. This seems like a delicate and important issue, but I found the description of how it is solved to be lacking. - Lines 70-74 are redundant. Perhaps one is an older draft that should have been deleted? - For readers coming from the (non-contextual) linear bandit setting, achieving log(T) regret as opposed to sqrt(T) regret may be a bit surprising. Perhaps you can add a sentence in the related work section as to why this is expected and reasonable beyond directly citing previous work. - In the contributions, you list an implementation detail, noting that the estimation subroutine need not be run to completion, but only until a certain property is satisfied (eq. 2, presumably?). To what extent do you think this is a theoretical artifact versus an actual prescription for someone wishing to implement this? - To what extent is assuming knowledge of a gap common and reasonable? In other bandit setting this can have strong impact on the guarantees. I guess in this case, assuming a smaller gap, would simply lead to more exploration early on. - As a matter of interest, to what extent do you think the regret bound is tight? In the matrix completion literature, the sample complexity can scale like O(r(k +d)). Where does the extra factor of r come from in your bound and is it tight? - NeurIPS seems like a kind of a strange setting for a paper of this length compared to COLT, for instance. I feel like by the end of the main body, we really only have the algorithm set up, and then the analysis follows a blitzkrieg of assumptions and a quick theorem statement which makes the reading strange. _________________update___________________ Thank you for taking the time to address my concerns and answer the questions. I appreciated the simulation given in the author response and think it would make a valuable addition to the paper. I maintain that the structure could do with some rearrangement if possible to address how compressed section 4 is, and hope to see a longer form of this released somewhere as well.

Reviewer 2



This submission studies the MAB in high dimension case with the low-rank structure and proposes an algorithm with a scale of regret r^2*(k+d). Comments: -add related references (1-3) whose cluster size is equivalent to the rank and sharing the same spirit which should be discussed at least, and (1) is also like a way in alternating minimization. References: (1) Collaborative Filtering Bandits, (2) On Context-Dependent Clustering of Bandits, (3) Distributed Clustering of Linear Bandits in Peer to Peer Networks -add a section to describe to your main assumptions and motivate them -add a section to articular what's the main theoretical difference with (6,15) as well as suggested advances -section 3 needs to significantly shrink to save some spaces for the experimental part which is also crucial to the success of this draft -section 4 seems incomplete, add a proof sketch would be sufficient for readers to get the main view -change your name from REAL to RELR, etc, that better represents proposed algorithm -move two-phase bandit algorithm from appendix to main text -add an experimental section to verify the proposed claims -add an explanation regarding lemma 5 and 6 and their relationship Overall, I went through the whole submission, and found that your theoretical contribution is expected, though needs a better organization and polish, whereas currently your empirical contribution is none, which is hard to go through at this venue, this you are highly encouraged to provide, and there are some jobs need to be done to improve the quality of this draft, in short, my recommendation is this submission with current content is slightly below the acceptance bar, but I am open to change my score when I see their response. %%%updated comments I read all comments and the response from the authors, my concern has been addressed, and my overall recommendation is an acceptance.

Reviewer 3



it seems that REAL-estimator is an post-processing after standard low-rank optimization. If so I didn't quite get what's the significance in the technical improvement. The paper is clearly written but not compact. I have lists a few confusing/unclear points in the improvement section.

[Author Response · NeurIPS 2019]



We thank all reviewers for their valuable suggestions on citing relevant literature and improving
organization of the paper. Responses to other questions are presented below.

**Simulations.** Reviewer 1 found the paper more on the theoretical side and also asked about practi-
cality of our theory-driven algorithm, and Reviewer 2 requested a numerical study. We compared
REAL Bandit with 4 other algorithms: OLS-Bandit of [15][1], LASSO-Bandit of [6], OFUL of [2]
which is UCB based, and Thompson sampling (from [36]). We generated matrix $\mathbf{B}^\star$ as $\mathbf{U}\mathbf{V}^\top$ where
$\mathbf{U} \in \mathbb{R}^{201\times 3}$ and $\mathbf{V} \in \mathbb{R}^{200\times 3}$ with iid $\mathcal{N}(0,1)$ entries. Noise variance is 1 and features are iid
$\mathcal{N}(\mathbf{0}, \mathbf{I}_d)$. We gave Thompson sampling the true prior mean and variance of arm parameters, and true
noise variance. We generated 10 data sets, ran all algorithms, and present their average cumulative
regret (with 1 SE error bars) for a time horizon of length $T = 40,000$ in the above figure. We are
grateful to the reviewers for this suggestion since the simulations back our theoretical results.

**Reviewer 1.** *Practicality of the gap assumption:* unlike [2] or [36], we do not assume a deterministic
gap exists (this would actually not hold since covariates can be very close to the decision boundaries).
Our Assumption 3 of §A (adapted from [6,15]) only requires that a subset of arms are optimal with
positive probability, and the remaining arms are sub-optimal with positive probability. It can be
shown this assumption holds for all standard distributions for the covariates. *Optimality of factor $r^2$*
*in the regret:* we thank the reviewer for this, since it led us to a careful investigation of the bounds
which made us realize the bounds can actually be tightened to replace $r^2$ with $r$, matching the bounds
one sees in matrix completion literature.

**Reviewer 2.** *Theoretical contributions beyond [6,15]:* we highlight these major advances: 1) We
provide a stronger characteristic of "all-sampling" observations which helps obtaining tail-bound
inequalities for the all-sampling estimator (see function $G$ in Assumption 12 of §C). The analysis
in [6,15] are not sufficient for our low-rank estimator bound. 2) We allow for a randomized forced-
sampling rule which is more flexible in practice than the deterministic sampling rules introduced in
[6, 15]. 3) We proposed a simple method for identifying the optimal arms which is required for the
analysis of estimators to work. *Explanation of Lemmas 5-6 and their relationship:* Lemmas 6-7 verify
the assumptions of row-enhancement bounds while Lemma 5 verifies those of the trace-regression
estimator. The former is concerned about the samples for *each individual* arm, whereas the latter does
not care about individual arms and only demands iid samples among $\mathbf{X}_i$'s (which include samples
from multiple arms). However, they are very similar in nature.

**Reviewer 3.** *Significance of the REAL-estimator:* low-rank tail bounds are known for $\|\widehat{\mathbf{B}} - \mathbf{B}^\star\|_F$,
but we need row-wise bounds $\|\widehat{B}_\kappa - B_\kappa^\star\|_2$. One can use $\|\widehat{B}_\kappa - B_\kappa^\star\|_2 \leq \|\widehat{\mathbf{B}} - \mathbf{B}^\star\|_F$, but this is
loose by a factor $\sqrt{k}$. REAL-estimator allows to avoid this by showing that the error in $\widehat{\mathbf{B}} - \mathbf{B}^\star$
is spread roughly equally among different rows. *Regularity assumptions on context vectors:* we
note that Assumptions 1-3 also appear in [6] and [15], and [6] discusseses and demonstrates their
practical relevance on real data. Assumption 4 requires the data not to be heavy-tailed and explore all
directions. Assumption 5 is concerned about the accuracy of low-rank estimator which is standard in
low-rank matrix estimation literature.

## Footnotes

[1]We adopt the same numbers for the references as the submission.


[Meta-Review · NeurIPS 2019]

Reviewers found the paper well-prepared and thoroughly argued. The rebuttal clarifying the related work and the simulation appears to have satisfied the reviewers, moving some scores.